# Immune Cells Invade the Collateral Circulation during Human Stroke: Prospective Replication and Extension

**DOI:** 10.3390/ijms22179161

**Published:** 2021-08-25

**Authors:** Marc Strinitz, Mirko Pham, Alexander G. März, Jörn Feick, Franziska Weidner, Marius L. Vogt, Fabian Essig, Hermann Neugebauer, Guido Stoll, Michael K. Schuhmann, Alexander M. Kollikowski

**Affiliations:** 1Department of Neuroradiology, University Hospital of Würzburg, 97080 Würzburg, Germany; Strinitz_M@ukw.de (M.S.); Pham_M@ukw.de (M.P.); Maerz_A@ukw.de (A.G.M.); Feick_J@ukw.de (J.F.); Weidner_F1@ukw.de (F.W.); Vogt_M2@ukw.de (M.L.V.); 2Department of Neurology, University Hospital of Würzburg, 97080 Würzburg, Germany; Essig_F@ukw.de (F.E.); Neugebauer_H@ukw.de (H.N.); Stoll_G@ukw.de (G.S.); Schuhmann_M@ukw.de (M.K.S.)

**Keywords:** ischemic stroke, cerebral ischemia, mechanical thrombectomy, large vessel occlusion, leukocytes, neutrophils, collateral circulation

## Abstract

It remains unclear if principal components of the local cerebral stroke immune response can be reliably and reproducibly observed in patients with acute large-vessel-occlusion (LVO) stroke. We prospectively studied a large independent cohort of *n* = 318 consecutive LVO stroke patients undergoing mechanical thrombectomy during which cerebral blood samples from within the occluded anterior circulation and systemic control samples from the ipsilateral cervical internal carotid artery were obtained. An extensive protocol was applied to homogenize the patient cohort and to standardize the procedural steps of endovascular sample collection, sample processing, and laboratory analyses. N = 58 patients met all inclusion criteria. (1) Mean total leukocyte counts were significantly higher within the occluded ischemic cerebral vasculature (I) vs. intraindividual systemic controls (S): +9.6%, I: 8114/µL ± 529 vs. S: 7406/µL ± 468, *p* = 0.0125. (2) This increase was driven by neutrophils: +12.1%, I: 7197/µL ± 510 vs. S: 6420/µL ± 438, *p* = 0.0022. Leukocyte influx was associated with (3) reduced retrograde collateral flow (R^2^ = 0.09696, *p* = 0.0373) and (4) greater infarct extent (R^2^ = 0.08382, *p* = 0.032). Despite LVO, leukocytes invade the occluded territory via retrograde collateral pathways early during ischemia, likely compromising cerebral hemodynamics and tissue integrity. This inflammatory response can be reliably observed in human stroke by harvesting immune cells from the occluded cerebral vascular compartment.

## 1. Introduction

The cumulative incidence of cerebrovascular diseases is considerable in high-income countries (218 (95% CI 214–221) per 100,000 in men and 127 (95% CI 125–128) per 100,000 in women, respectively) and has reached epidemic levels in low to middle-income countries [1,2]. Among these diseases, acute ischemic stroke (AIS) is the leading cause of death and disability, which, despite recent major advances in acute stroke therapy, prompts the urgent need for adjunct treatments beyond macrovascular recanalization [3,4]. In AIS, the immune system exerts a strong and early inflammatory response, which is directed at the ischemic brain [5,6]. Immune cell infiltration into the ischemic brain represents a central component of this response [7], which has been widely and repeatedly observed in the most commonly practiced experimental models of ischemic stroke across species and over decades of stroke research [8,9]. However, detailed spatiotemporal information on immune cell infiltration, including its route of traffic, is particularly lacking for the early phase after vascular occlusion and for the human system in general [5,10]. There is ample evidence from murine intraluminal filament occlusion models closely mimicking human large-vessel-occlusion (LVO) stroke that the modulation of stroke inflammation may be such a promising adjunct therapeutic target [5,11]. Importantly, in experimental settings, the (post) stroke inflammatory response seems to emerge from the cerebral vascular compartment [5]. In acute human stroke, thus far, only sites located remotely from ischemia have been accessible to observation, e.g., by retrieving samples of peripheral blood or of lumbar cerebrospinal fluid (CSF) [12]. Consequently, the inability of human cerebral observation of stroke-induced inflammation in close temporal and anatomical proximity to the ischemic event remains an obstacle for translation [13,14,15,16]. Only recently, the ischemic cerebral vasculature of acute stroke patients has become accessible for investigation during mechanical thrombectomy (MT), while, importantly, recanalization still has not taken place: We and few other groups established the method of blood aspiration from within the ischemic cerebral vasculature during MT, which is performed with a distally placed microcatheter at the end of the occlusion phase immediately before therapeutic recanalization is achieved more proximally through stent-embolus-retrieval [17,18,19]. In principle, this method can expose local pathophysiology for scientific observation if patient-related, interventional, and laboratory confounders are controlled by protocol [17,20,21]. However, the reproducibility and consistency of observations made by this approach remain unclear. A recent study used flow cytometry to define relative immune cell populations within the occluded cerebral vasculature of AIS patients but did not report on absolute immune cell numbers [14]. Consequently, our human observation of local neutrophil infiltration could not be reproduced thus far [17]. The present study aims to (1) prospectively assess local immune cell infiltration quantitatively in a large second independent cohort of LVO stroke patients by applying an extensive protocol, which homogenizes the study population and standardizes the procedural steps for quality control. Furthermore, (2) the impact of local immune cell infiltration on hemodynamic-functional (collateral flow) and radiological-structural (infarct extent) parameters is investigated.

## 2. Results

### 2.1. Patient Characteristics

N = 116 samples of *n* = 58 consecutive LVO stroke patients meeting all pre-specified protocol criteria (58 local cerebral-ischemic (I) vs. 58 systemic control samples (S)) entered data analyses. Demographic, clinical, radiological, therapy- and sampling-related patient characteristics are shown in Table 1. Briefly, the mean patient age was 74 ± 11 years. Women were slightly overrepresented (66%). Hypertension was the most prevalent comorbid disease (90%), followed by atrial fibrillation (60%). Median clinical stroke severity as measured by the National Institutes of Health Stroke Scale (NIHSS) at hospital admission was 15 (10–19). Median baseline infarct extent as assessed by the pre-interventional Alberta Stroke Program Early CT score (ASPECTS) was 8 (7–9). N = 24 patients (41%) received intravenous thrombolysis before endovascular therapy for LVO. Median time from symptom onset to groin puncture was 255 (190–325) min. Initial cerebral angiograms revealed *n* = 38 (66%) middle cerebral artery (MCA) M1 occlusions, *n* = 17 (29%) proximal M2 occlusions, and *n* = 10 (17%) internal carotid artery (ICA-T) occlusions, respectively. Median time from symptom onset to local cerebral-ischemic sampling was 295 (231–295) min. The mean time delay of retrograde collateral blood flow (relative time to peak opacification, rTTP) to reach the target site of cerebral-ischemic blood sampling was 3.8 ± 1.4 s. Final cerebral angiograms revealed technically successful recanalization (mTICI ≥ 2b) in *n* = 43 patients (83%) after a median onset-to-final-recanalization time of 335 (253–381) min. Mean duration of antegrade blood flow to reach the mid-insular MCA target site after recanalization was 1.3 ± 0.6 s. Median follow-up ASPECTS after recanalization therapy was 7 (6–8). Median clinical stroke severity as assessed by NIHSS at 72 h after MT was 13 (4–18). Parenchymatous hematoma (> 30% of infarcted tissue) following MT occurred in *n* = 2 patients (3%). N = 4 (7%) patients succumbed during the hospital stay. Detailed results of baseline characteristics are displayed in Table 1.

### 2.2. Cerebral Immune Cell Recruitment

The total number of leukocytes in the local cerebral-ischemic (I) samples obtained under occlusive conditions was significantly increased compared with the systemic (S) control samples obtained from the cervical ICA (+9.6%, I: 8114/µL, 95% CI 7056 to 9173 vs. S: 7406/µL, 95% CI 6469 to 8344, *p* = 0.0125). This local increase in leukocyte counts within the occluded vascular compartment was primarily driven by a significant increase in the neutrophil subpopulation (+12.1%, I: 7197/µL, 95% CI 6175 to 8220 vs. S: 6420/µL, 95% CI 5541 to 7298, *p* = 0.0022). There was neither a significant difference in the local total number of leukocytes (−0.4% men, I: 8091/µL, 95% CI 6272 to 9911 vs. women: 8126/µL, 95% CI 6763 to 9490, *p* = 0.9751) nor a significant difference in the local neutrophil (−4.3%, men, I: 6994/µL, 95% CI 5303 to 8684 vs. women: 7308/µL, 95% CI 5963 to 8653, *p* = 0.7716), lymphocyte (+12.5%, men, I: 694/µL, 95% CI 502 to 886 vs. women: 617/µL, 95% CI 464 to 769, *p* = 0.3426) and monocyte (−4.4%, men, I: 130/µL, 95% CI 85 to 175 vs. women: 136/µL, 95% CI 80 to 193, *p* = 0.5197) cell count between men and women. Likewise, there were no significant differences between the control sample and local cerebral-ischemic lymphocyte (−8.3%, I: 644/µL, 95% CI 527 to 760 vs. S: 702/µL, 95% CI 548 to 856, *p* = 0.8127) or monocyte counts (−8.8%, I: 134/µL, 95% CI 95 to 173 vs. S: 147/µL, 95% CI 96 to 198, *p* = 0.6341). Total and differential leukocyte counts are shown for each sampling location in Figure 1.

### 2.3. Association of Cerebral Immune Cell Recruitment with Local Blood Flow

Linear regression was performed to investigate the impact of the observed immune cell influx on retrograde collateral flow under occlusive conditions and on antegrade blood flow following recanalization (rTTP). Local cerebral-ischemic total leukocyte (β = 0.0001027, R^2^ = 0.09696, *p* = 0.0373) and lymphocyte counts (β = 0.001095, R^2^ = 0.1082, *p* = 0.0313) were significantly correlated with the collateral transit time to reach the sampling site located distally to the occlusive target lesion, translating into an increase of collateral transit time of 0.5 s for every increase of 4869 total leukocyte counts/µL. Local cerebral-ischemic neutrophil and monocyte counts were also positively associated with rTTP, however, without statistical significance. None of the local cell counts were correlated with antegrade blood flow measured after the restoration of antegrade flow immediately after recanalization. The results of these analyses are given in Figure 2.

### 2.4. Association of Cerebral Immune Cell Recruitment with Infarct Extent

Associations of cerebral-ischemic immune cell recruitment with baseline and follow-up infarct extent as assessed by ASPECTS are shown in Figure 3. Local cerebral-ischemic total leukocyte counts were correlated with reduced baseline (β = −0.0001149, R^2^ = 0.08382, *p* = 0.032) and follow-up (β = −0.0001299, R^2^ = 0.07097, *p* = 0.0472) ASPECTS, translating into a decrease of 1 baseline ASPECTS point for every increase of 8211 total leukocyte counts/µL. Local neutrophil counts were also correlated with infarct extent prior to recanalization (β = −0.0001218, R^2^ = 0.08878, *p* = 0.0356). None of the other differential leukocyte counts was associated with infarct extent on baseline or follow-up imaging.

### 2.5. Immune Cell Profiles in AIS Patients with Cervical ICA Tandem Lesions

We additionally and independently analyzed immune cell counts in the subgroup of patients (*n* = 23) with tandem lesions at the cervical ICA level (i.e., stenosis > 50%, percutaneous transluminal angioplasty, stenting, and ipsilateral dissection). These data are given in Figure 4. Intraindividual analyses of these patients (i.e., immune cell counts from ICA lesion level vs. cerebral-ischemic immune cell counts) revealed a shift towards higher total and differential leukocyte counts at the ICA lesion level, which was statistically significant for monocytes (+29.3%, ICA lesion: 128/µL, 95% CI 74 to 182 vs. I: 99/µL, 95% CI 60 to 138, *p* = 0.0426).

## 3. Discussion

AIS elicits a strong innate inflammatory response, which emerges from the vascular compartment and which is directed at the ischemic brain region [22,23]. As the dominant constituents of this response, (1) local immune cell infiltration, (2) platelet activation, and (3) the interaction between infiltrating immune cells and activated platelets could be identified predominantly in experimental stroke models [5,11,24]. It would represent significant progress for translational stroke research to observe parts of these responses reliably and reproducibly in a more direct manner within the human cerebral vasculature [15,16]. In the pursuit of this aim, few groups, including ours, were able to establish a novel method by which blood samples can be obtained with a microcatheter from within the ischemic territory during MT for acute LVO stroke [17,18,19]. With this method, immune cells can be harvested directly from the occluded cerebral vascular compartment immediately before it is reopened by stent-embolus-retrieval. In other words, such local observation was not disturbed due to non-ischemic forward (antegrade) blood flow because sample aspiration was performed under strictly occlusive conditions [17]. As opposed to myocardial infarction, this unique opportunity has emerged in LVO stroke because the culprit lesion, in the vast majority of cases, is represented by an embolus, which can be penetrated with a microcatheter without concomitant vessel recanalization [17,25,26].

In previous work, we established an extensive prospective protocol defining the flow of consecutive patients to homogenize the target population [17]. In addition, clinical-radiological, angiographic-interventional, and sampling-related variables were highly standardized. Importantly, protocol composition was guided by closely considering current concepts of the stroke-related immune response and vascular inflammation [5,11,15,27,28,29]. A central aspect in this regard was the control standard to which samples from the occluded cerebral vascular compartment were compared. To achieve a local (cerebral arterial), inter-individual control standard remains elusive because invasive intervention and blood aspiration from corresponding cerebrovascular sites is not possible in non-stroke control subjects for ethical reasons [30]. To alternatively achieve a systemic (peripheral venous) inter-individual control standard for immune cell counts also remains elusive because such independent controls are inherently not comparable to acute inflammatory responses in the arterial compartment of the brain vasculature [31]. In addition, there is a substantial variation of systemic immune cell counts even between healthy individuals [32]. Sample sizes of smaller dimensions due to a complex study protocol, including many variables of prospective patient flow, sample aspiration, sample processing, and laboratory analysis, cannot overcome such a degree of variation [17,32]. For these reasons, one must argue for an intra-individual control standard of comparison. To be able to test and re-test the specific hypothesis of cerebral immune cell infiltration in the human system, intra-individual conditions where vascular inflammation and cell activation at a given systemic control level could confound local observation at the cerebrovascular target level need to be excluded by protocol.

There is evidence that atherosclerotic plaques within ICA stenoses upon inflammation-driven rupture per se activate platelets and the coagulation system, leading to cervical thrombus formation as a major cause of embolic ICA and MCA occlusions [33]. In line with these findings, elevated levels of neutrophil extracellular traps (NETs), activated platelets, and markers of platelet-derived microparticles have been reported at the site of cervical ICA occlusions in AIS patients [27]. To directly address this issue, we additionally analyzed immune cell counts in local cerebral-ischemic and carotid artery samples in an independent cohort of patients with ICA stenosis or dissection who were excluded from the primary cohort a priori. In this group of AIS patients, analyses in fact revealed numerically increased levels of immune cells at the site of cervical ICA tandem lesions. These data lend further empirical support to exclude any vascular lesion site as control for cerebral observation during AIS from a priori since they represent a confounding source of vascular inflammation themselves. In addition, several other protocol variables probably are also critical to becoming able to observe immune cell infiltration into the occluded cerebral vascular compartment. For example, in those LVO lesions where occlusion is not total but sub-occlusive (corresponding to mTICI 1), minimal residual forward (antegrade) blood flow passes into the ischemic compartment and may disturb local observation by contamination.

Thus far, we were able to prospectively and consecutively observe the two largest independent cohorts (*n* = 318 in this study, *n* = 183 previously reported [17]) in which immune cells of AIS patients (*n* = 58 in this study, *n* = 40 previously reported [17]) could be harvested from the occluded cerebral vascular compartment and analyzed with a high degree of standardization [34]. As a main finding of the current study, local immune cell infiltration into the occluded cerebral vascular compartment could be reproduced with very similar qualitative and quantitative observations in comparison to previous observations [17]. Specifically, it was replicated that neutrophils represent the dominant infiltrating immune cell population. In addition, the quantitative estimate of local vs. systemic neutrophil count was reproduced with considerable accuracy: Current replication of +12.1% (*p* = 0.0022) vs. previous finding of +13.1% (*p* = 0.0018) [17]. Since the literature suggests women have higher numbers of total leukocytes and neutrophil counts and have higher rates of overall complications, in-hospital deaths, and worse functional following AIS, our main finding of cerebral immune cell recruitment during AIS was additionally tested for sex differences in local immune cell counts which, however, could not be shown [35,36].

We are led to believe that the human observation of local neutrophil infiltration into the occluded cerebral vascular compartment represents a true and consistent finding and no scientific or statistical error [15,16]. This finding is also highly plausible from a biological perspective within the experimental framework of ischemia-related inflammation: neutrophils are known to be among the first immune cells to invade the ischemic brain [28]. Furthermore, replication was possible in a second and independent cohort using identical a priori defined criteria of inclusion/exclusion without missing or selecting observations. The interpretation of our results may be further extended. Although we cannot directly prove the route and time course of neutrophil traffic, we may safely assume that, during complete LVO, vascular routes through leptomeningeal collateral (vascular) channels represent the major route of neutrophil infiltration at least during the very early phase after onset of cerebral ischemia [37]. This notion is fuelled by the experimental facts that neutrophils do not locally reside within but enter the brain only after pathological stimuli such as, e.g., ischemic stroke [28]. Further supporting this interpretation, regression analysis revealed the association of cerebral immune cell recruitment during occlusive conditions with dynamic (retrograde) blood flow (rTTP) and structural infarct extent (ASPECTS). These measures suggest that patients affected by slow (poor) collateral flow as well as those with more extensive structural damage show locally higher total leukocyte and particularly neutrophil density. Observations of this kind might reflect downstream effects of microvascular plugging and/or elevation of blood viscosity, which in turn could reduce retrograde collateral flow by increasing vascular resistance [9,38].

In conclusion, the observation of neutrophil-dominant immune cell recruitment into the occluded cerebral vasculature of AIS patients could be prospectively and accurately reproduced in a large cohort by applying a highly standardized protocol as proposed by our group. Since we repeatedly detected this finding in a short-lived immune cell subpopulation, we recommend using this protocol for quality assurance of endovascular sampling, sample processing, and sample preparation. Future therapeutic breakthroughs of stroke immune modulation may depend on these novel methods of harvesting human immune cells directly, locally, and acutely during a stroke. In this manner, in-depth human immune cell phenotyping is becoming possible for reverse translation.

## 4. Materials and Methods

### 4.1. Study Design

This investigation was designed as a prospective cross-sectional study of consecutive ischemic stroke patients who underwent emergency MT due to acute symptomatic occlusion of the distal ICA, MCA M1, and proximal M2 segment between July 2019 to November 2020. Briefly, we analyzed cerebral arterial blood obtained by microcatheter aspiration from within the ischemic cerebral vasculature, as previously reported by our group [17,20,21]. Ethical approval was obtained from the ethics committee of the University of Würzburg (#135/17). All patients or their legal representatives provided written informed consent.

### 4.2. Inclusion Criteria

Patient inclusion criteria for the prospective assessment of local cerebral immune cell infiltration during AIS were defined as follows: (1) acute ischemic stroke (AIS) with severe neurological baseline deficit qualifying for mechanical thrombectomy (MT) according to current guidelines [3] and (2) periprocedural (invasive angiographic) confirmation of LVO of the following sites: distal ICA (ICA-T), MCA M1 segment or proximal M2 segment, respectively. Patients were excluded for the following reasons: (1) proven bilateral or multifocal LVO other than defined; (2) angiographically proven residual or restored antegrade blood flow [29]; (3) any deviation from the interventional, sampling, and pre-processing protocol; (4) LVO in conjunction with either ≥50% cervical ICA stenosis or ICA dissection; and (5) intraprocedural percutaneous transluminal angioplasty (PTA) or stent implantation [27].

Patient inclusion criteria for the prospective assessment of immune cell profiles in the subgroup of AIS patients with cervical ICA tandem lesions were defined as follows: (1) AIS with severe neurological baseline deficit qualifying for MT according to current guidelines [3]; (2) invasive angiographic confirmation of LVO of the following sites: distal ICA (ICA-T), MCA M1 segment or proximal M2 segment; and (3) LVO in conjunction with either ≥ 50% cervical ICA stenosis or ICA dissection; and (4) intraprocedural percutaneous transluminal angioplasty (PTA) or stent implantation [27]. Patients were excluded for the following reasons: (1) proven bilateral or multifocal LVO other than defined; (2) angiographically proven residual or restored antegrade blood flow [29]; (3) any deviation from the interventional, sampling, and pre-processing protocol.

### 4.3. Interventional and Sampling Protocol

Indication for MT by stent-embolus retrieval was given by acute symptomatic LVO [3]. Endovascular access for mechanical thrombectomy was obtained by a transfemoral approach using the modified Seldinger technique. Recanalization of the predefined target lesions was preceded by microcatheter navigation (Neuroslider 27 or 21; Acandis, Pforzheim, Germany) into the mid-insular middle cerebral artery M2 segment. The choice of a distinct microcatheter was at the discretion of the operator for reasons of variable vascular anatomy. After microcatheter positioning and discarding the respective microcatheter-specific dead space volume, a sample of 1 mL of local cerebral blood (ischemic) was drawn with 3 mL luer-lock syringe (Omnifix, B.Braun, Melsungen, Germany) if possible [17]. Samples obtained from the ipsilateral ICA were used as controls (systemic) and drawn analogously after clot removal and/or termination of the thrombectomy procedure [20,21]. All samples were immediately transferred into citrate–phosphate–dextrose–adenine (CPDA-1) monovettes (S-Monovette; Sarstedt, Nümbrecht, Germany) for anticoagulation.

### 4.4. Radiological Parameters

Pre-interventional infarct extent at stroke center presentation and early follow-up infarct extent at 24–48 h following recanalization were measured through the Alberta Stroke Program Early CT Score (ASPECTS; CT: Somatom Definition AS; Siemens Healthineers, Erlangen, Germany) [39]. Post-processing algorithms (syngo iFLOW software; Siemens Healthineers, Erlangen, Germany) were applied for parametric color coding of digital subtraction angiography (DSA) series and for the quantification of collateralization during occlusive conditions. The retrograde collateral flow was assessed by means of collateral transit time as defined by relative time to peak opacification (rTTP). rTTP was determined by using a region-of-interest (ROI) analysis, where predetermined ROI locations reflected the site of local cerebral sampling and the mid-insular target region receiving retrograde blood supply under occlusion conditions. rTTP was calculated by subtracting the time to peak opacification (TTP) of ROI 1 (circular, placed at the lacerum segment of the internal carotid artery, measuring 10.80 mm²) from the TTP of ROI 2 (circular, placed centered at the mid-insular sampling location, measuring 309.60 mm²) [40]. The angiographic degree of final recanalization following MT was graded by the modified treatment in cerebral infarction scale (mTICI) [41]. Antegrade macrovascular blood flow was assessed by means of rTTP at the site of local cerebral sampling following final recanalization.

### 4.5. Laboratory Analysis

A time window ≤ 12 h after completion of sampling was allowed for sample processing. All laboratory analyses were performed by trained personnel unaware of clinical data. The CPDA-1-anticoagulated blood was used for cell counting and the preparation of blood smears. After red blood cell lysis and white blood cell (WBC) staining with Tuerk solution (Merck, Darmstadt, Germany), total WBC counts were determined by using a Fuchs Rosenthal hemocytometer. Thin blood smears were prepared from 5 μL of whole blood (glass slides: R. Langenbrinck, Emmendingen, Germany). Blood smears were stained by standard Pappenheim stain (Merck). Cell counts of neutrophils, lymphocytes, and monocytes were calculated by multiplying the percentages in Pappenheim-stained blood smears with the respective WBC counts.

### 4.6. Prospective Patient Flow

The total eligible cohort comprised *n* = 318 patients who presented with suspected LVO between July 2019 and November 2020. A flowchart of patient inclusion is provided in Figure 5.

Of these *n* = 318 patients, *n* = 43 were excluded for vertebrobasilar occlusion sites, and *n* = 14 patients for bilateral or multifocal occlusions other than the ipsilateral ICA or MCA. Of the remaining *n* = 261 patients, *n* = 53 patients failed to meet study inclusion criteria either because of angiographically proven sub-occlusion with residual antegrade blood flow or complete recanalization. Intraprocedural microcatheter aspiration of local cerebral-ischemic blood samples was attempted in *n* = 208 patients. According to the study protocol, precise occlusion locations were angiographically proven and limited to the following sites of anterior circulation LVO: ICA-T, MCA M1, and proximal M2 segment. Local microcatheter aspiration was attempted in all *n* = 208 patients but remained empty (sicca) in *n* = 83/208 patients. It succeeded in *n* = 125/208 patients (60%). Of these *n* = 125 patients, *n* = 19 patients were excluded for protocol violations of the sampling technique or sampling order (e.g., inadvertent reversal of sampling order local vs. systemic, non-insular target location, omission of second/control sample). N = 23 patients were excluded because of the relevant tandem lesion at cervical level (stenosis > 50%) with or without subsequent percutaneous transluminal angioplasty (PTA) and/or stenting, also including cases with primary or secondary dissection of ipsilateral cervical ICA [42,43]. These cervical tandem lesions were considered as potentially powerful confounders from a priori since local vascular inflammation should be expected at these cervical sites [27]. N = 25 patients could not be included for reasons of poor sample quality or of deviation from the sample pre-processing protocol (insufficient sample volume or time delay > 12 h between sampling and sample analysis). At the end of consecutive patient flow, *n* = 58 patients met all a priori defined clinical-radiological, sampling, interventional, and laboratory criteria of study inclusion.

### 4.7. Statistical Analysis

Statistical analyses were performed using GraphPad Prism (GraphPad Prism 9.0.0, GraphPad Software, San Diego, CA, USA) and MedCalc 19.6 (MedCalc Software, Ostend, Belgium). The normal distribution of datasets was tested using the D’Agostino-Pearsons test. Data are given as mean with 95% confidence interval (CI), as mean ± standard deviation (SD), as median with interquartile range (IQR), or absolute and relative frequency distribution unless otherwise specified. The paired t-test or Wilcoxon Rank-Sum test was used to compare cell counts of related samples. The unpaired t-test or Mann–Whitney U test was used to compare cell counts of unrelated samples. Simple linear regression was performed to identify associations between cerebral-ischemic cell counts, hemodynamic-functional, and radiological-structural parameters. All *p* values reported were 2-sided with *p* < 0.05 being considered statistically significant.

### 4.8. Manuscript Preparation

The manuscript was prepared according to the STROBE (Strengthening the Reporting of Observational Studies in Epidemiology) statement for observational studies [44].

## Figures and Tables

**Figure 1 ijms-22-09161-f001:**
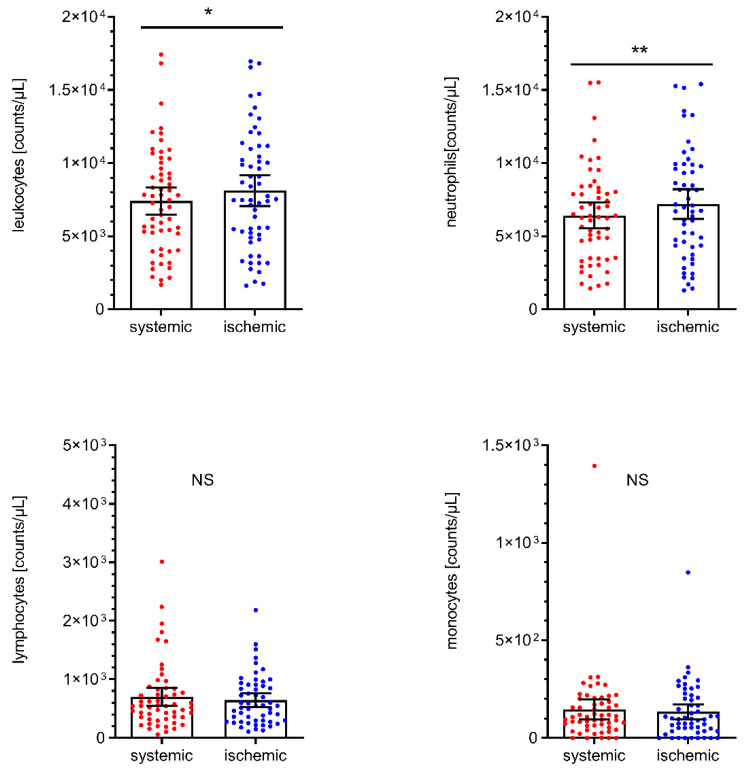
Total leukocyte, neutrophil, lymphocyte, and monocyte cell counts by sampling location. Scatter dot plot with mean and 95% confidence interval (CI). Total leukocyte cell counts (upper left panel) were tested by paired T-test. ** *p* < 0.005, *n* = 58. Differential leukocyte cell counts were tested by the Wilcoxon rank-sum test. * *p* < 0.05. NS, not significant.

**Figure 2 ijms-22-09161-f002:**
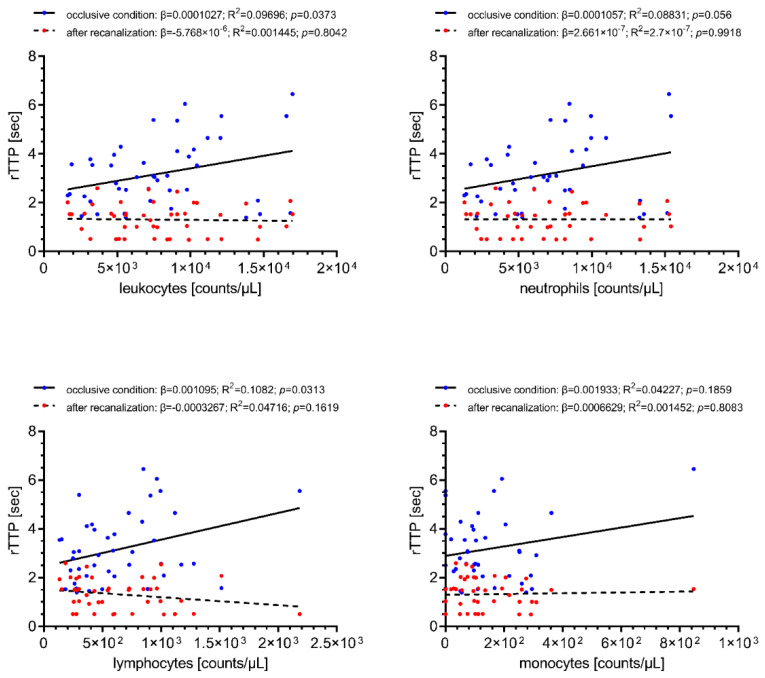
Simple linear regression of local immune cell counts with collateral transit time during occlusive conditions (blue) and antegrade blood flow (red) following recanalization (rTTP). Increased cerebral-ischemic total leukocyte and lymphocyte counts are associated with longer collateral transit time. Total leukocyte counts, *n* = 57; differential leukocyte counts, *n* = 52. rTTP, relative time to peak opacification; R^2^, coefficient of determination; β, regression coefficient.

**Figure 3 ijms-22-09161-f003:**
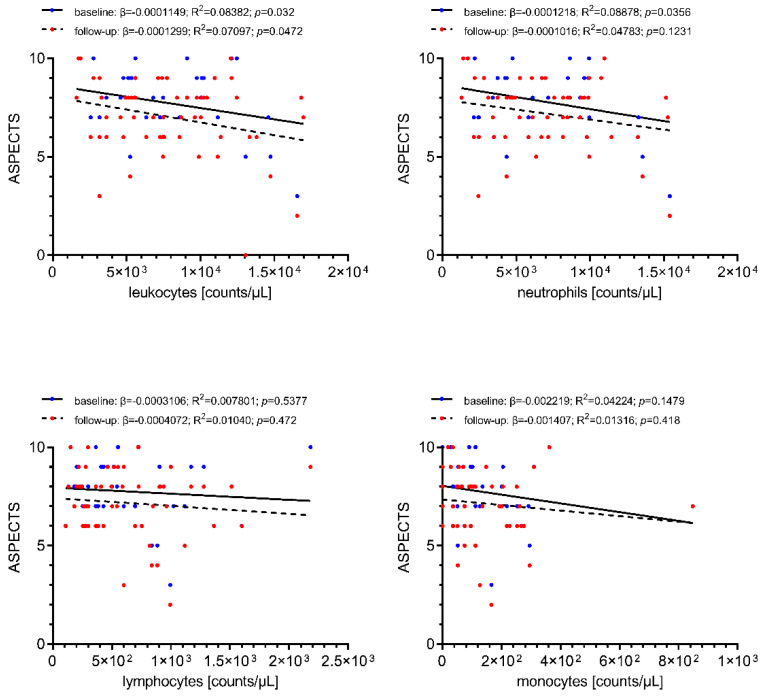
Simple linear regression of local immune cell counts with pre-interventional infarct extent at stroke center presentation (blue) and early follow-up infarct extent (ASPECTS) at 24–48 h following recanalization (red). Increased cerebral-ischemic total leukocyte and neutrophil counts are correlated with reduced baseline and follow-up ASPECTS (statistically significant only for total leukocyte counts). Total leukocyte counts, *n* = 58; differential leukocyte counts, *n* = 54. ASPECTS, Alberta Stroke Program Early CT score; R^2^, coefficient of determination; β, regression coefficient.

**Figure 4 ijms-22-09161-f004:**
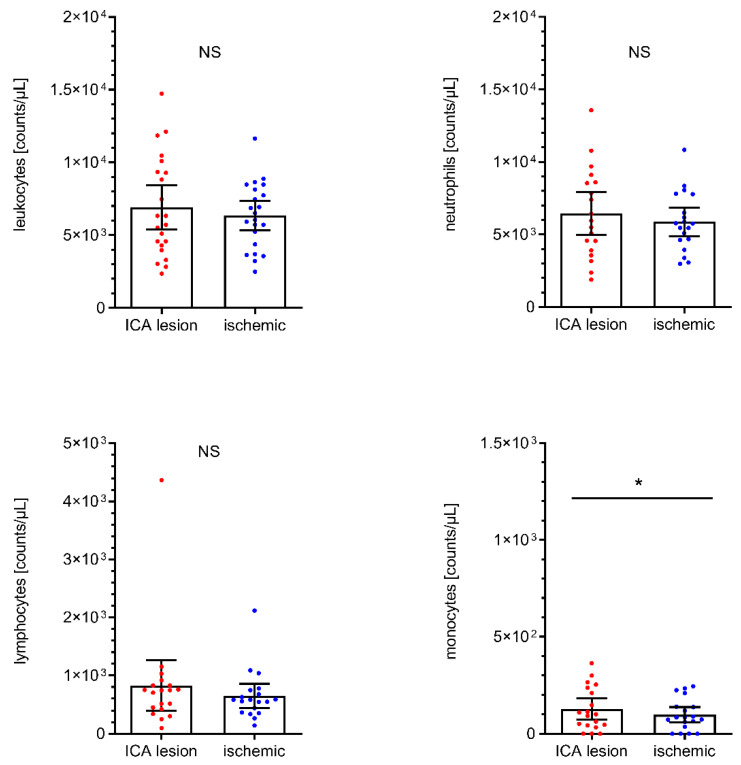
Analyses of total leukocyte (*n* = 22), neutrophil (*n* = 19), lymphocyte (*n* = 19) and monocyte (*n* = 19) cell counts in AIS patients with tandem lesion at ipsilateral cervical ICA level (including stenosis > 50%, transluminal angioplasty/stenting, and ICA dissection). Scatter dot plot with mean and 95% confidence interval (CI). Total leukocyte cell, neutrophil, and monocyte counts were tested by paired T-test. Lymphocyte counts were tested by the Wilcoxon rank-sum test. * *p* < 0.05. NS, not significant. ICA, internal carotid artery.

**Figure 5 ijms-22-09161-f005:**
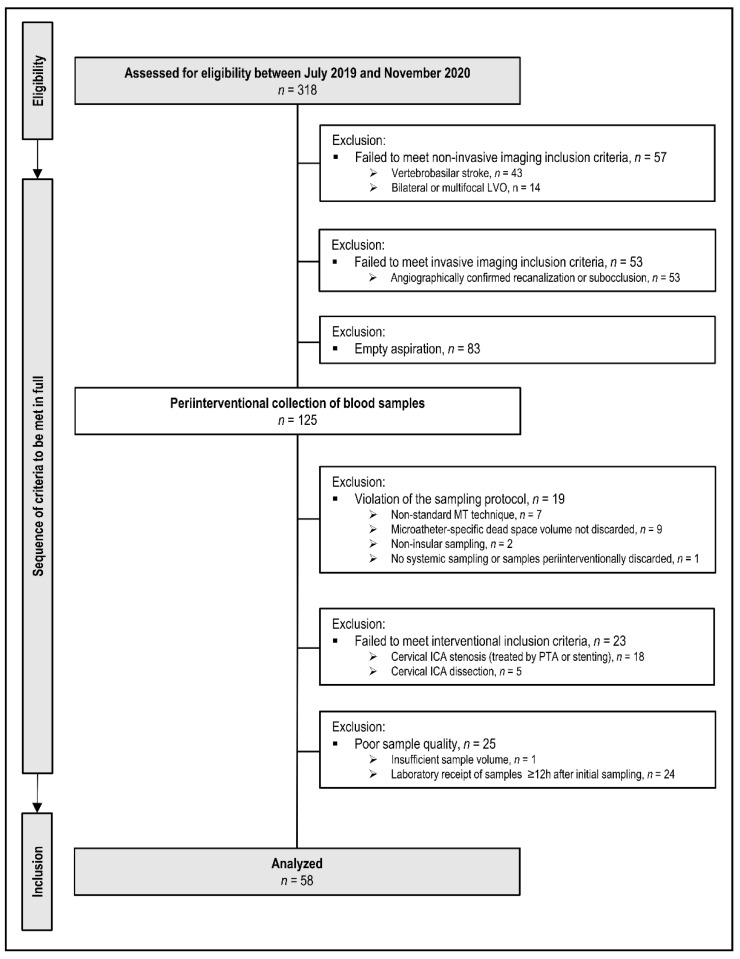
Flow chart of patient inclusion and exclusion according to the prospective protocol (multiple exclusion criteria possible). Non-standard abbreviations: ICA, internal carotid artery; LVO, large-vessel-occlusion; MT, mechanical thrombectomy; PTA, percutaneous transluminal angioplasty.

**Table 1 ijms-22-09161-t001:** Characteristics of included patients.

Demographics	
Age, years (SD)	74 (11)
Male sex, *n* (%)	20 (34)
**Medical history**	
Hypertension, *n* (%)	52 (90)
Diabetes mellitus, *n* (%)	14 (24)
Hyperlipidemia, *n* (%)	21 (36)
Atrial fibrillation, *n* (%)	35 (60)
Current smoker, *n* (%)	9 (16)
Baseline medication	
Antithrombotic medication, *n* (%)	30 (52)
Antihypertensive drugs, *n* (%)	52 (90)
**Clinical presentation**	
Systolic blood pressure, mmHg (IQR)	158 (140–178)
Diastolic blood pressure, mmHg (IQR)	80 (67–97)
Heart rate, min^−1^ (SD)	86 (23)
NIHSS at presentation (IQR)	15 (10–18)
Unknown time of symptom onset, *n* (%)	19 (33)
ASPECTS at presentation (IQR)	8 (7–9)
**Treatment**	
IV rt-PA, *n* (%)	24 (41)
Intervention	
Onset-to-puncture, min (IQR)	255 (190–325)
Angiographic occlusion location *	
M1, *n* (%)	38 (66)
M2, *n* (%)	17 (29)
ICA, *n* (%)	10 (17)
Tandem occlusion, *n* (%)	5 (9)
rTTp (occlusive condition), s (SD)	3.2 (1.4)
Successful recanalization, *n* (%)	48 (83)
Onset-to-final-recanalization, min (IQR)	335 (253–381)
rTTp (after recanalization), s (SD)	1.3 (0.6)
Sampling	
Onset-to-distal sampling, min (IQR)	295 (231–295)
Onset-to-carotid sampling, min (IQR)	352 (275–414)
**Outcome**	
ASPECTS post-intervention (IQR)	7 (6–8)
NIHSS 72 h post intervention (IQR)	13 (4–18)
Parenchymatous hematoma, *n* (%)	2 (3)
In-house mortality, *n* (%)	4 (7)
Neurological cause of death, *n* (%)	3 (5)
Non-neurological cause of death, *n* (%)	1 (2)

Categorical data are presented as number (*n*) with percentage (%). Continuous data are presented as mean with standard deviation (SD) or median with interquartile range (IQR). Non-standard abbreviations: ASPECTS, Alberta Stroke Program Early CT score; ICA, internal carotid artery; IV rt-PA, intravenous recombinant tissue plasminogen activator; M1/M2, middle cerebral artery section; min, minutes; mmHg, millimeters of mercury; NIHSS, National Institutes of Health Stroke Scale; rTTP, relative time to peak opacification; s, seconds. * Including multiple sites per patient.

## Data Availability

The datasets generated during and/or analyzed during the current study are available from the corresponding author on reasonable request.

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
