# Peer review of "Immune Cells Invade the Collateral Circulation during Human Stroke: Prospective Replication and Extension"

_ijms, 2021, doi:10.3390/ijms22179161_

Round 1
Reviewer 1 Report
The authors quantitatively examined local immune cell infiltration in a prospective cohort of patients with large vessel occlusion (LVO) stroke by applying an extensive protocol that homogenizes the study population and standardizes procedural steps for quality control. They analyzed the impact of local immune cell infiltration on hemodynamic-functional (collateral flow) and radiological-structural (infarct extent) parameters. Blood samples could be obtained with a microcatheter from within the ischemic territory during mechanical thrombectomy (MT) for acute LVO stroke. The authors found that mean total leukocyte count was significantly higher within the occluded ischemic cerebral vasculature. Specifically, neutrophils represented the dominant infiltrating immune cell population, and leukocyte influx was associated with reduced retrograde collateral flow and greater infarct extent. The study is potentially interesting, but can be improved if the following considerations are addressed:
1. In order to highlight the epidemiological importance of cerebrovascular diseases, it would be useful to mention in the Introduction the results of an epidemiologic study in Catalonia (Spain) on acute stroke (Rev Esp Cardiol 2007; 60; 573-580). In this study, the cumulative incidence of cerebrovascular diseases per 100,000 population was 218 (95% CI, 214-221) in men and 127 (95% CI, 125-128) in women.
2. It is mandatory to describe the causes of death (neurological and non-neurological) in the study sample.
3.Did the authors find differences in their results with respect to gender? A recently published study on the impact of female gender on the distribution of risk factors, stroke subtype, stroke severity, and outcome (Clin Neurol Neurosurg 2014 Dec;127:19-24) in acute ischemic stroke should also be discussed in the Discussion.
Author Response
Point-by-Point reply
ID: ijms-1338147
"Immune cells invade the collateral circulation during human stroke:
Prospective replication and extension”
REVIEWER #1:
The authors quantitatively examined local immune cell infiltration in a prospective cohort of patients with large vessel occlusion (LVO) stroke by applying an extensive protocol that homogenizes the study population and standardizes procedural steps for quality control. They analyzed the impact of local immune cell infiltration on hemodynamic-functional (collateral flow) and radiological-structural (infarct extent) parameters. Blood samples could be obtained with a microcatheter from within the ischemic territory during mechanical thrombectomy (MT) for acute LVO stroke. The authors found that mean total leukocyte count was significantly higher within the occluded ischemic cerebral vasculature. Specifically, neutrophils represented the dominant infiltrating immune cell population, and leukocyte influx was associated with reduced retrograde collateral flow and greater infarct extent. The study is potentially interesting, but can be improved if the following considerations are addressed:
Point #1: In order to highlight the epidemiological importance of cerebrovascular diseases, it would be useful to mention in the Introduction the results of an epidemiologic study in Catalonia (Spain) on acute stroke (Rev Esp Cardiol 2007; 60; 573-580). In this study, the cumulative incidence of cerebrovascular diseases per 100,000 population was 218 (95% CI, 214-221) in men and 127 (95% CI, 125-128) in women.
Our answer: We thank the reviewer for these encouraging remarks and the specific suggestion to incorporate additional literature on the general epidemiological importance of cerebrovascular diseases. The literature provided was cited in detail as suggested (page 1, lines 32 to 35).
Corresponding changes:
MAIN TEXT
INTRODUCTION
“The cumulative incidence of cerebrovascular diseases is considerable in high-income countries (218 (95% CI 214-221) per 100000 in men and 127 (95% CI 125-128) per 100000 in women, respectively) and has reached epidemic levels in low to middle income countries [1,2]. Among these diseases, acute ischemic stroke (AIS) is the leading cause of death and disability which, despite recent major advances in acute stroke therapy, prompts the urgent need for adjunct treatments beyond macrovascular recanalization. [3,4]. […]”
Point #2: It is mandatory to describe the causes of death (neurological and non-neurological) in the study sample.
Our answer: This is a good suggestion. The causes of death were further specified and presented in table 1. As suggested, a distinction was made between neurological (n=3) and non-neurological (n=1) causes of death (page 3 and 4).
Corresponding changes:
MAIN TEXT
RESULTS
Table 1. Characteristics of included patients.
“[…]
Outcome |
|
[…] |
|
In-house mortality, n (%) |
4 (7) |
Neurological cause of death, n (%) |
3 (5) |
Non-neurological cause of death, n (%) |
1 (2) |
[…]”
Point #3: Did the authors find differences in their results with respect to gender? A recently published study on the impact of female gender on the distribution of risk factors, stroke subtype, stroke severity, and outcome (Clin Neurol Neurosurg 2014 Dec;127:19-24) in acute ischemic stroke should also be discussed in the Discussion.
Our answer: To comprehensively address this point, we have additionally analyzed all local cerebral-ischemic immune cell counts for their potential difference between men and women. There was neither a significant difference in the local total number of leukocytes (-0.4% men, I: 8091/µl, 95%CI 6272 to 9911 vs women: 8126/µl, 95%CI 6763 to 9490, p=0.9751) nor a significant difference in the local neutrophil (-4,3%, men, I: 6994/µl, 95%CI 5303 to 8684 vs women: 7308/µl, 95%CI 5963 to 8653, p=0.7716), lymphocyte (+12,5%, men, I: 694/µl, 95%CI 502 to 886 vs women: 617/µl, 95%CI 464 to 769, p=0.3426) and monocyte (-4.4%, men, I: 130/µl, 95%CI 85 to 175 vs women: 136/µl, 95%CI 80 to 193, p=0.5197) cell count. These data were incorporated into the results section (page 4, lines 113 to 119) and discussed against the background of the literature provided by the reviewer (page 8, lines 243 to 247).
Corresponding changes:
MAIN TEXT
RESULTS
“[…] There was neither a significant difference in the local total number of leukocytes (-0.4% men, I: 8091/µl, 95%CI 6272 to 9911 vs women: 8126/µl, 95%CI 6763 to 9490, p=0.9751) nor a significant difference in the local neutrophil (-4,3%, men, I: 6994/µl, 95%CI 5303 to 8684 vs women: 7308/µl, 95%CI 5963 to 8653, p=0.7716), lymphocyte (+12,5%, men, I: 694/µl, 95%CI 502 to 886 vs women: 617/µl, 95%CI 464 to 769, p=0.3426) and monocyte (-5.1%, men, I: 130/µl, 95%CI 85 to 175 vs women: 137/µl, 95%CI 80 to 193, p=0.5197) cell count between men and women.[…]”
DISCUSSION
“[…] Since literature suggests women to have higher numbers of total leukocytes and neutrophil counts and to have higher rates of overall complications, in-hospital deaths, and worse functional following AIS, our main finding of cerebral immune cell recruitment during AIS was additionally tested for sex differences in local immune cell counts which, however, could not be shown [35,36].
We are led to believe that the human observation of local neutrophil infiltration into the occluded cerebral vascular compartment represents a true consistent finding and no scientific or statistical error [15,16]. […]”

Reviewer 2 Report
In my opinion, the current layout of the article is not appropriate. The Materials and methods section from lines 290 to 376 should be moved after the introduction. Also, I don't quite understand why the part about the specific exclusion criteria for patients is included in the results. For me, this is confusing, because in the end 58 patients were analysed, not 318. I would suggest putting the part from line 67-98 in the materials and methods section.
Line 122 - Not all abbreviations are explained in the description below the table - incl. SD, IQR - please explain all abbreviations. Moreover, abbreviations in the description below the table should be described in alphabetical order.
Line 155 - there is an editorial error here - this is the end of the sentence above the graphs.
Author Response
Point-by-Point reply
ID: ijms-1338147
"Immune cells invade the collateral circulation during human stroke:
Prospective replication and extension”
REVIEWER #2:
Point #1: In my opinion, the current layout of the article is not appropriate. The Materials and methods section from lines 290 to 376 should be moved after the introduction.
Our answer: Thank you for the attentive reading of our manuscript. We have thoroughly rechecked the manuscript for any layout issues and cross-checked the journal’s layout requirements. According to the journal’s requirements, the materials and methods section must follow the discussion (https://www.mdpi.com/journal/ijms/instructions).
Point #2: Also, I don't quite understand why the part about the specific exclusion criteria for patients is included in the results. For me, this is confusing, because in the end 58 patients were analysed, not 318. I would suggest putting the part from line 67-98 in the materials and methods section.
Our answer: This is a reasonable suggestion. To preserve clarity throughout the manuscript, the passages associated with specific exclusion criteria were incorporated into the material and methods section (page 11, line 353 to page 12, line 384).
Point #3: Line 122 - Not all abbreviations are explained in the description below the table - incl. SD, IQR - please explain all abbreviations. Moreover, abbreviations in the description below the table should be described in alphabetical order.
Our answer: All abbreviations are now explained in the legend (page 4, lines 99 to 105). Abbreviations standing alone without accompanying explanatory text are presented in alphabetical order as suggested. This has also been applied to the legend of figure 5 (page 11, lines 358 to 361).
Corresponding changes:
MAIN TEXT
RESULTS
Table 1. Characteristics of included patients.
“[…] Categorial data are presented as number (n) with percentage (%). Continuous data are presented as mean with standard deviation (SD) or median with interquartile range (IQR).
Non-standard abbreviations: ASPECTS, Alberta Stroke Program Early CT score; ICA, internal carotid artery; IV rt-PA, intravenous recombinant tissue plasminogen activator; M1/M2, middle cerebral artery section; min, minutes; mmHg, millimeters of mercury; NIHSS, National Institutes of Health Stroke Scale; rTTP, relative time to peak opacification; sec, seconds.
*Including multiple sites per patient. […]”
MATERIALS AND METHODS
“[…] Figure 5. Flow chart of patient inclusion and exclusion according to the prospective protocol (multiple exclusion criteria possible). Non-standard abbreviations: ICA, internal carotid artery; LVO, large-vessel-occlusion; MT, mechanical thrombectomy; PTA, percutaneous transluminal angioplasty. […]”
Point #4: Line 155 - there is an editorial error here - this is the end of the sentence above the graphs.
Our answer: The editorial error has been corrected as suggested.
